# Longitudinal Association of Physical Activity, Mastery and Psychological Distress in Mid-Aged Adults over 9-Years

**DOI:** 10.3390/ijerph192114052

**Published:** 2022-10-28

**Authors:** Adam J. Novic, Charrlotte Seib, Nicola W. Burton

**Affiliations:** 1School of Applied Psychology, Griffith University, Brisbane 4122, Australia; 2Centre for Mental Health, Griffith University, Brisbane 4122, Australia; 3Menzies Health Institute Queensland, Griffith University, Brisbane 4222, Australia; 4School of Nursing and Midwifery, Griffith University, Brisbane 4215, Australia

**Keywords:** mental health, well-being, self-control, exercise, mid-aged adults

## Abstract

Psychological distress is highly prevalent and associated with significant adverse health outcomes and economic burden. Mastery and physical activity are potential resources to reduce distress and promote wellbeing; however, previous research has not examined their potential interactive relationship over time. The purpose of this study was to explore associations between mastery, physical activity, and distress in mid-aged adults over nine years. Data from a longitudinal mail survey study including the Kessler 6, Pearlin Mastery Scale, and items assessing time spent in physical activity were examined in a sample of 4404 adults aged 40 to 54 years at baseline. Group-Based Trajectory Models identified two distinct trajectories of psychological distress (elevated and low). Generalized Estimating Equations were used to assess mastery and physical activity, adjusting for sociodemographic and health variables, as predictors for the probability of distress group membership. The odds of elevated distress over time were significantly reduced in people with higher mastery (OR = 0.13; 95% 0.11–0.15) and doing at least 150 min/week of physical activity (OR = 0.81; 95% 0.68–0.96). There was no significant interaction between mastery and physical activity. Mastery and physical activity may be important resources to mitigate distress and further research is needed to evaluate interventions promoting these resources and the impact on mid-aged adults experiencing psychological distress.

## 1. Introduction

Psychological distress is highly prevalent among mid-aged adults [1,2,3], and is significantly associated with adverse health outcomes and economic burden. At the individual level, psychological distress has a strong positive association with anxio-depressive disorders and increases the odds of mid-aged adults accessing mental health services [4,5]. Psychological distress increases the risk of major chronic diseases, such as arthritis, cardiovascular disease, chronic obstructive pulmonary disorder; and has a positive dose–response relationship with all-cause mortality [6,7]. At the macro level, psychological distress increases absenteeism resulting in an estimated AUD$5.9 billion in relation to reduced productivity [8], and costs around USD$3364 per capita in direct healthcare [9].

The severity of psychological distress is influenced by the utility of available resources. The ability to cope with life stressors is considered fundamental to safeguarding against psychological harm and subsequent dysfunction [10,11]. Within this framework, psychological resources and specific coping behaviors are considered as modifiable factors operating as protective mechanisms [12], though more research is needed to identify associations with the long-term patterns of psychological distress.

Mastery is a psychological resource that is conceptualized as the extent one perceives a sense of control over important life outcomes and is a predictor of emotional wellbeing in mid-aged adults [11,13]. A sense of mastery in mid-aged adulthood has an inverse association with negative affect and a positive association with life satisfaction and positive affect [14]. It has been identified as a strong negative predictor of anxiety and depression for people in early mid-aged adulthood and has been suggested to enable coping through engagement with social supports [15,16].

Physical activity is also inversely associated with incident psychological distress [17]. Among adults who are physically inactive throughout the week, there is an increased risk of reduced psychosocial wellbeing [18], whereas increases in the frequency of physical activity have been shown to reduce the odds of experiencing psychological distress in adults [19,20]. For mid-aged adults, increased engagement in physical activity has been shown to significantly reduce the likelihood of reporting serious psychological distress [21].

Inter-relations between physical activity and mastery on psychological distress have also been observed. Mastery has been shown to buffer the relationship between physical activity and psychological distress such that variations in distress are more distinct between physical activity levels when mastery is low; those who are inactive and have low mastery are the most likely to experience distress than any other combination of mastery and physical activity [22]. Engagement in physical activity has also been found to indirectly influence the relationship between mastery and psychological health, suggesting that those with a greater sense of control are more likely to engage in physical activity and thereby maintain or increase psychological health [23]. Although these studies have suggested a relationship between mastery and physical activity, they have not examined the potential interaction with psychological distress over time.

The aim of this study was to explore associations between physical activity, mastery and distress in mid-aged adults over time. Mid aged adults were focused on given this age group has the highest proportion of people across the lifespan reporting very high levels of distress [1]. The results of this research can be used to inform the development of evidence-based interventions to manage psychological distress in mid-aged adults.

## 2. Materials and Methods

This study presents a secondary data analysis of selected participants in the HABITAT study, a multi-level five-wave longitudinal study (i.e., 2007 to 2016) investigating physical activity, as well as psychological, social, and environmental factors, and select health and wellbeing outcomes in mid-aged and older men and women. The HABITAT survey was awarded initial ethical clearance by the QUT Human Research Ethics Committee in 2005 (ID3967H).

The design and recruitment process of HABITAT have been described in detail elsewhere [24,25]. Briefly, HABITAT employed a multi-stage probability sampling design to obtain a stratified random sample of 200 neighbourhoods from the Australian Bureau of Statistics Census Collection Districts (CCDs), which were ranked by the Index of Relative Socioeconomic Disadvantage, an indicator reflecting attributes such as proportions of residents with low income, low educational attainment, and workers in relatively unskilled occupations. Data from the Australian Electoral Commission were used within each of the selected neighbourhoods to identify an average of 85 households where there was at least one individual aged 40 to 65 years (as of March 2007). One person from each household was then randomly selected and invited to participate (i.e., 17,000 adults). Each person was assigned a unique identification code which was printed on their questionnaire to enable matching across survey waves.

A structured self-report questionnaire was administered using a mail-survey method adapted from Dillman [26], with questionnaires personalised to the local area and age range of the participants. Advanced personalised notice of delivery and one-week reminders/thank you for completion and return were also provided via mail. Those who did not respond within seven weeks of the initial mailout were sent a replacement questionnaire. The first survey wave commenced in 2007 (n = 11,035, 68.43% response rate) and the last in 2016 (n = 5187, 58.77% response rate). When compared with 2006 census data, the 2007 HABITAT sample was broadly representative of the wider Australian mid-aged population [27].

### 2.1. Sample

The current study used data from all five waves of HABITAT with an analytical sample of 4404 respondents aged 40 to 54 years at baseline. This ensured that participants remained in the mid-age bracket at all available time points, i.e., 2009, 2011, 2013, and 2016. Analyses included those who had data collected at two or more waves. Figure 1 presents the flowchart of the derived analytic sample and participant attrition throughout the study period. Longitudinal inconsistencies in the dataset (e.g., changes in gender from baseline) were reviewed with 191 cases excluded due to suspected change in respondent across waves. All participants were sent the questionnaire at each timepoint (even if there was no response previously) with the exception of participants who were deceased, and those who advised that they were not interested in study participation, unavailable (e.g., travelling overseas) or incapable due to a physical or mental condition.

### 2.2. Measures

#### 2.2.1. Psychological Distress

Psychological distress was measured using the Kessler 6 (K6) scale [28]. The six-item instrument assesses how often respondents experienced anxio-depressive symptoms in the past month and uses a 5-point Likert scale with responses ranging from 0 (none of the time) to 4 (all of the time). Items are summed using a standard scoring algorithm [29], with total scores ranging from 0 to 24 with higher scores indicating more psychological distress. The K6 has previously been shown to have a Test Information Curve (TIC) value of 34 to 54 with precision of scale scores between 0.14 and 0.24 standard errors in the target severity range among Australian adults aged 18 years and over [28], indicating that changes in observed scores in that severity range strongly relate to changes in true scores. The K6 has excellent internal consistency in adults aged 18 and over (*α* = 0.92, [28]).

#### 2.2.2. Physical Activity

Physical activity was assessed using items from the Active Australia Survey (AAS) [30] which assessed walking (including to get to and from places and for exercise/recreation), moderate physical activity (e.g., gentle swimming, social tennis, golf), and vigorous physical activity (e.g., jogging, cycling, aerobics, competitive tennis). Respondents report the duration (hours and minutes) of each category of physical activity in the last week. The AAS items exhibit good reliability and acceptable validity [30,31] and have been recommended as suitable for use in Australian population-based research [31].

In accordance with reporting guidelines, total time in physical activity was a weighted calculation computed by adding time spent in walking, moderate physical activity, and two times the amount spent in vigorous physical activity given the higher intensity, and truncated to avoid errors due to over-reporting [30]. To aid interpretability, physical activity was categorized according to weekly weighted minutes: (1) less than 60 min, (2) 60 to 149 min, (3) 150 to 300 min, (4) over 300 min. Groups were selected to ensure sensitivity to quantities of physical activity below and above global recommendations of at least 150 min per week [32].

#### 2.2.3. Mastery

Mastery was assessed using the Pearlin Mastery Scale (PMS) [11], a 7-item scale that measures the extent to which an individual generally feels as though they maintain a sense of control over important life outcomes. Respondents are asked to indicate to what extent they agree with items using a 5-point Likert scale ranging from 1 ‘strongly disagree’ to 5 ‘strongly agree’. Items are summed resulting in a total score ranging from 7 to 35 with higher scores indicating a greater mastery. Scores of mastery were dichotomised based on a one third vs. two third split to accommodate the negatively skewed distribution: (1) Low, score of 25 or less, (2) High, score over 25.

#### 2.2.4. Covariates

Variables identified from previous research as covariates of psychological distress were also considered in this analysis [12,33,34,35]. They included demographic variables (age, gender, and highest educational qualification) and self-reported general health (rated as poor/fair, good, or very good/excellent).

### 2.3. Statistical Analyses

All data were analyzed using the Statistical Package for the Social Science (SPSS) version 28 [36] or STATA version 17 [37] statistical packages. The distributions of variables were graphically inspected using frequency distributions and normal P-P plots and were summarized as means (standard deviation [SD]) and proportions (number, percentage). The level of statistical significance was set at α = 0.05.

Group-based trajectory modelling (GBTM) was conducted within STATA using the TRAJ command to identify heterogenous psychological distress trajectory subgroups over time [38]. A two-stage model selection process was used to identify the optimum number of trajectory groups whilst also considering the description of raw data, model parsimony, and theoretical relevance. The adequacy of the distinctly different longitudinal trajectories was tested using the initial null model for comparison. The final number of groups and polynomial order was determined by examining difference between model Bayesian Information Criterion (BIC) via an approximate Bayes factor (B_ij_), obtaining close correspondence between the estimated probability of group membership and the proportion assigned to the group, average posterior probability of subgroup membership for the assigned model with acceptability set to ≥0.7 (values close to 1 indicating higher precision), odds of correct classification based on posterior probabilities of group membership ≥ 5, narrow confidence intervals around group membership probabilities, and the proportion of participants in each group [39,40,41,42]. Chi-square tests of homogeneity were conducted for each sociodemographic variable to examine the distributions between the two distress level groups identified by the GBTM analyses.

Generalized estimating equations (GEE) method’s logistic regression with an independent working correlation structure and robust standard errors was used to predict the probability of distress trajectory group membership [43]. The choice of this model was made following two considerations: first, the outcome was derived from trajectory modelling that resulted in independent groupings across the study period. Second, the model fit of correlation structures were examined and compared using the Quasi-likelihood Information Criteria (QIC), with lower values indicating better model fit. An interaction term was included in the initial model to assess differences between categories of mastery and physical activity on the trajectory of psychological distress and precision of estimates was measured using 95% confidence intervals (CI).

## 3. Results

### 3.1. Missing Data Analysis

The proportion of missing data for psychological distress was 9.3% at survey wave 2, 10.3% at wave 3, 15.2% at wave 4, and 30.6% at wave 5. Similar proportions of missingness were also observed for mastery (9.4%, 10.6%, 15.0%, 30.5%) and physical activity (9.4%, 9.9%, 14.7%, 31.0%). Missing data for time spent in physical activity (i.e., walking, moderate, and vigorous intensity) were managed according to standard guidelines [30] such that cases with all missing data were deleted and cases which included data for at least one category of physical activity had zero (0) imputed if any of the other categories of physical activity had missing data. Missing data analyses indicated data to be missing completely at random (Little’s MCAR, χ^2^(212) = 236.602, *p* = 0.118).

### 3.2. Main Analyses

Group membership of psychological distress trajectories across mid-aged adulthood was the primary outcome. Prior to trajectory modelling, a sensitivity analysis was performed to assess the robustness and consistency of variations in psychological distress by survey wave participation. The average score of psychological distress in the sample varied significantly across study wave (χ^2^(3) = 22.92, *p* < 0.001). Respondents at wave 3 reported higher scores of distress than for all other waves, and no differences in distress were indicated between remaining study waves. Across study waves, 64.2% of participants reported at least 150 weighted minutes per week of physical activity which is comparable with international recommendations for good health [32]. There was an increasing proportion of participants reporting low mastery over time (χ^2^(3) = 11.37, *p* = 0.010). Levels of psychological distress revealed no significant difference between patterns of wave completion (χ^2^(14) = 22.75, *p* = 0.064), therefore data were pooled for the GBTM.

GBTM was conducted in two stages to determine best model fit. First, the number of groups was determined using BIC and average posterior probability (see Table 1), while the three-group model showed slightly improved BIC values, the addition of groups over two resulted in a reduction in the average posterior probability, larger standard errors of group membership estimates, and did not provide meaningful interpretation of scores of psychological distress (e.g., 3-group model contained two groups persistently below a score of 5 on the K6, indicating likely not distressed). Therefore, the two-group model was considered the best fit considering model parsimony and distinct trajectory patterns related to psychological distress, depicted in Figure 2 with 95% confidence intervals. The final model showed good overall fit (BIC (14,734) = −35,356.62; BIC (4399) = −35,351.18; Entropy = 0.86) and contained an adequate proportion of participants in each group: Group 1, comprising 81.2% of participants showed stable and generally low levels of distress (i.e., below a score of 5 on K6) whereas Group 2 (18.8%) comprised participants with stable and elevated psychological distress (Figure 3). Compared to the low distress group, those with elevated distress reported consistently higher levels across mid-age adulthood (M_difference_ = 6.60, SE = 0.27).

Summary characteristics of the two distress groups at baseline are presented in Table 2. Overall, the average age of the sample at baseline was 47 years (SD = 4.30), most were women (56.0%), born in Australia (78.1%), in the paid workforce (86.2%), and residing with a partner (73.2%). The elevated distress group contained a greater proportion of women (χ^2^(1) = 19.80, *p* < 0.001), who were living alone or single (χ^2^(5) = 158.47, *p* < 0.001), not in the paid workforce (χ^2^(1) = 181.94, *p* < 0.001), with primary or secondary school education (χ^2^(2) = 179.44, *p* < 0.001), and reporting fair health (χ^2^(2) = 1082.13, *p* < 0.001). A greater proportion of participants with elevated distress over time was also not meeting physical activity recommendations (χ^2^(3) = 262.66, *p* < 0.001) and reporting low mastery (χ^2^(1) = 2508.55, *p* < 0.001) (Table 3).

Binary logistic regression using odds ratio (OR) and 95% confidence intervals (CI) was performed to examine the association between distress over time in mid-age adulthood with weekly time spent in physical activity and level of mastery. Socio-demographic factors and general health were also added to the model as covariates. The interaction term for physical activity and mastery was excluded from the final model due to an non-significant effect (χ^2^(3) = 1.30, *p* = 0.73). Full details of the final model are presented in Table 4. The odds of elevated distress over time was higher for women than men (OR = 1.26; 95% 1.06–1.51). Conversely, the odds of elevated distress were lower for those reporting their health as either good (OR = 0.48; 95% 0.41–0.56) or excellent (OR = 0.24; 95% 0.20–0.30), and higher levels of education after school including certificate/diploma (OR = 0.79; 95% 0.64–0.98) and bachelor/postgraduate degree (OR = 0.71; 95% 0.58–0.87).

Relative to a low level of mastery, high mastery was shown to substantially reduce the odds of experiencing elevated psychological distress over time (χ^2^(1) = 884.38, *p* < 0.001; OR = 0.13, 95% 0.11–0.15). For time spent in physical activity, meeting physical activity recommendations (i.e., at least 150 weighted minutes per week of moderate-vigorous intensity physical activity) resulted in significantly reduced odds of elevated distress over time (χ^2^(3) = 9.73, *p* < 0.05; OR_150–300min_ = 0.81, 95% 0.68–0.96; OR_>300min_ = 0.75, 95% 0.62–0.90).

## 4. Discussion

Understanding longitudinal patterns of associations among psychological and behavioral factors and psychological distress can inform interventions to reduce related burden in mid-aged adults. The aim of this study was to explore associations between physical activity, mastery and distress in mid-aged adults nine years. This study provides new insight into patterns of distress over time during mid-aged adulthood and two specific associated modifiable influences that can be targeted for intervention. Results showed that mastery and time spent in physical activity were independently and negatively associated with elevated distress over the study period, though no interaction was found.

Our results showed two distinct trajectories of psychological distress in mid-aged adults over the nine-year study period: low and elevated distress that was generally stable over time. This is consistent with previous large-scale research regarding the stability of psychological distress in Australian adults [3]. The stability in elevated distress over time may be explained by associated factors which are also generally stable over time. For example, the use of maladaptive coping strategies (e.g., self-blame, avoidance) tends to be long-lasting [44], and could therefore contribute to long term distress. Enduring factors, such as physical illness, financial difficulties, and low level of education, may predispose individuals to chronic elevated levels of distress [33,35]. Consistent with this latter explanation, a greater proportion of people in the constantly elevated distress group in the current study had poor self-rated health, were not in paid employment, and had only secondary schooling. These characteristics are common components of low socio-economic status which has been shown to increase vulnerability to adverse mental health outcomes that perpetuate experiences of distress [45]. Our results also showed that a greater proportion of participants in the constantly elevated distress group were women, which is consistent with previous research showing that women are more likely than men to report high or very high levels of distress [34]. Earlier work has discussed women as more vulnerable to distress than men because of biological differences in stress reactivity and physiological responses, sex hormones, immune system differences; psychosocial differences related to cognitive style and behavioral responses; developmental processes; and sociocultural factors related to gender roles, increased vulnerability to trauma, and reduced access to resources [46,47]. Previous research has demonstrated links between social isolation and poor mental health outcomes and low recovery [48,49], and consistent with this, in the current study a greater proportion of participants in the constantly elevated distress group were living alone or single. Given these results, mid-aged adults with sociodemographic characteristics of residing alone or being single, unemployment, low education, and poor health, and who are also women may be priority target groups for interventions to ameliorate a trajectory of long-term distress. 

Mid aged adults with high mastery had a substantially lower odds of experiencing elevated psychological distress over the nine-year study period. This supports previous research demonstrating a protective effect of mastery on mental health. For example, other longitudinal research suggests increases in mastery are associated with decreases in psychological distress and increases in positive affect in mid-aged adults [13,50]. Mastery has been shown as a relatively stable resource in mid-aged adults and providing insulation from the impact of perceived life stressors, including financial problems, physical impairment, and poor health [12,51,52,53]. Notably, other research suggests that life stressors in mid-aged adults have a stronger deleterious impact on mastery than young adults [54].

Our results also showed that meeting physical activity guidelines of 150 min per week of moderate-vigorous intensity physical activity significantly reduced the odds of elevated distress in mid-aged adulthood over the nine-year study period. Other studies also provide support for a protective effect for those meeting physical activity guidelines [17,18,55]. Previous research has shown that physical activity may promote mental health through secretion of neurochemicals eliciting a sense of wellbeing [56], promoting self-esteem [57], and providing opportunity for positive social connections [58]. A previous longitudinal study demonstrated mental health benefits from exercise and sports only when done with others [59]. Physical activity done outside may provide exposure to urban greenness which has previously been inversely related to depressive symptoms [60] and positively related to psychological wellbeing [61]. There was, however, no significant relationship between lower levels of physical activity and psychological distress in the current study. This is contrary to previous dose–response studies finding reductions in psychological distress with small increases in physical activity to below recommended levels [19,20,21]. However, these previous studies measured physical activity *frequency*, and our study focused on physical activity *time*, which may in part explain differences in study results. More research is needed therefore on the “dose” of physical activity and the underlying mechanisms for good mental health.

Some methodological limitations of the current study are acknowledged. Measures of psychological distress, mastery, and physical activity used brief a referent period (e.g., in the last week) that may not reflect a typical experience. Participants may, for example, have done less physical activity than usual during the assessed period because of poor health, bad weather or other transient events. Assessing usual physical activity can however provide inflated estimates [31]. The survey used self-report data which are vulnerable to a social desirability bias such that respondents may answer questions in a way that would be perceived as favorable, and this might have changed the nature of associations. To aid interpretability of results, key predictor variables of mastery and physical activity were categorized, potentially resulting in loss of information and study power. Categories of mastery were determined in a meaningful way, but there are no recommended interpretable cut-offs for the Pearlin Mastery Scale [11] which was dichotomized according to obtained percentiles to minimize a negatively skewed distribution.

A major strength of this study is the longitudinal design covering a 9-year time frame with five waves of data, and a large sample size (*N* = 4404). This allowed for examination of patterns of distress over time and accounting for a range of baseline sociodemographic data. Statistically computing trajectory models of distress reduced risk of bias based on the use of pre-defined outcome categories, providing generalizable results specifically for mid-aged adults. Use of the Kessler scale as a well-established measure of psychological distress [28] aids in the interpretability and generalizability of results. We included well-known covariates of psychological distress in analyses, thereby reducing the risk of potential confounding.

## 5. Conclusions

This study provides new understanding of psychological distress and specific related cognitive and behavioral factors over time during mid-age adulthood. Distinct trajectories of low and elevated distress were identified in the sample and were relatively stable over time. Participants in the elevated distress group were more likely to be women, residing alone or single, not in paid employment, with school only education levels, and with poor self-rated health. These sociodemographic groups therefore represent priority targets for interventions. Both mastery and physical activity of at least 150 min per week were demonstrated to independently reduce the odds of sustained elevated distress, but did not interact. These two modifiable resources could therefore be included as separate components within psychotherapeutic interventions. Mastery may be enhanced, for example, through mindfulness-based strategies to promote psychological adaptability [62,63]. Physical activities, such as walking, may be included as a component of behavioral activation [64,65]. Applied research is needed to examine the feasibility and utility of interventions promoting mastery and physical activity and the impact on distress in mid-aged adults. Such interventions may help reduce associated adverse outcomes and economic burden.

## Figures and Tables

**Figure 1 ijerph-19-14052-f001:**
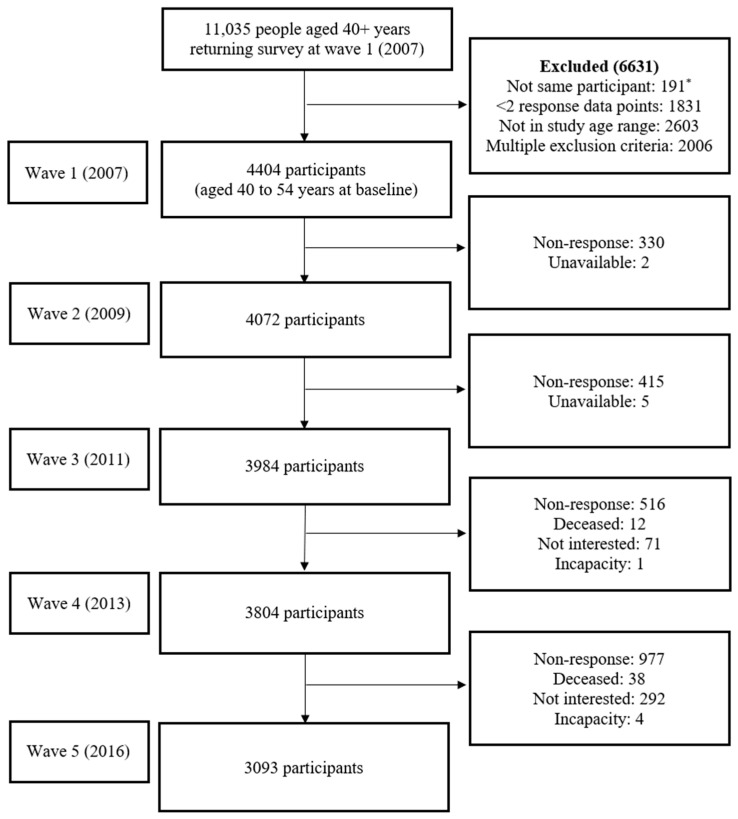
Flowchart of derived analytic sample of mid aged adults. * Longitudinal inconsistencies in the dataset (e.g., changes in gender) were assessed, and only those who were indicated to represent a single respondent across waves were included in analyses. Note: All participants were sent the questionnaire at each timepoint (even if there was no response previously) with the exception of participants who were deceased, and those who advised that they were not interested in study participation, unavailable (e.g., travelling overseas) or incapable due to a physical or mental condition. Non-response indicates the questionnaire was not returned or was returned without any data.

**Figure 2 ijerph-19-14052-f002:**
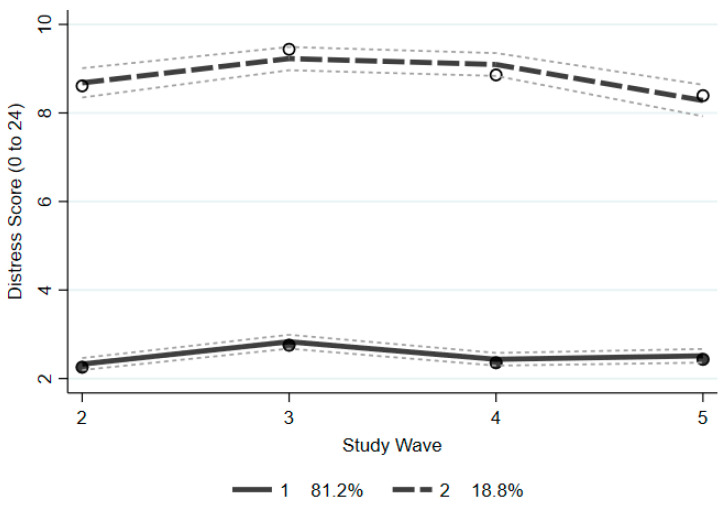
Trajectory patterns of psychological distress over four waves.

**Figure 3 ijerph-19-14052-f003:**
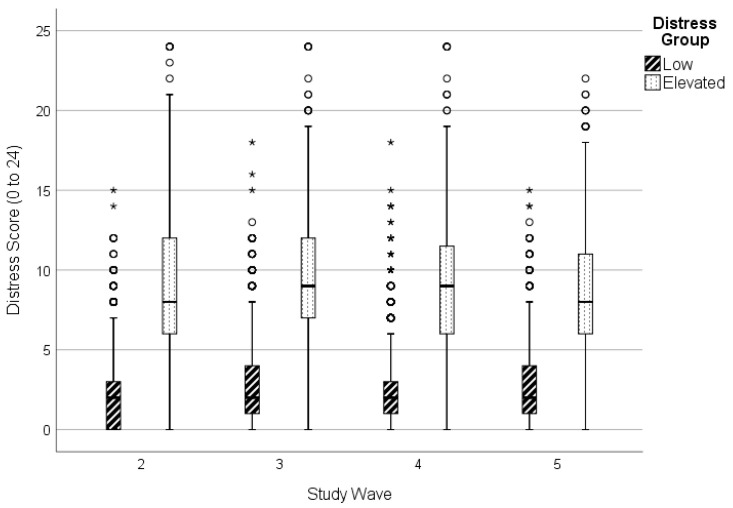
Psychological distress scores over time in low and elevated distress trajectory groups. ◦ Outlier beyond 1.5 times interquartile range. * Outlier beyond 3 times interquartile range.

**Table 1 ijerph-19-14052-t001:** Model search process for two-group psychological distress trajectories.

	Function	Function	Function	Function	Function
Group 1	Linear	Quadratic *	Quadratic *	Cubic *	Cubic *
Group 2	Linear	Quadratic *	Cubic *	Quadratic *	Cubic *
Entropy	0.856	0.855	0.853	0.857	0.855
BIC (N = 4399)	−35,386.72	−35,369.48	−35,368.01	−35,351.18 ^1^	−35,350.98 ^2^

* Function *p*-value < 0.01. ^1^, ^2^ Weak evidence to favor either model according to Jeffrey’s scale of evidence for Bayes factors [41].

**Table 2 ijerph-19-14052-t002:** Summary characteristics of analytic sample of mid aged adults at baseline (n = 4404).

	Total n (%) ^1^	Distress Level n (%)		*p* Value ^4^
		Low	Elevated	
Gender				<0.001
Men	1936 (44.0)	1613 (44.7)	323 (40.4)	
Women	2468 (56.0)	1992 (55.3)	476 (59.6)	
Country of Birth				0.002
Australia	3441 (78.1)	2,834 (78.6)	607 (76.0)	
Other	946 (21.5)	758 (21.0)	188 (23.5)	
Education				<0.001
School (year 12)	1146 (32.8)	1107 (30.7)	339 (42.4)	
Certificate/Diploma	1308 (29.7)	1084 (30.1)	224 (28.0)	
Bachelor/Postgraduate Degree	1640 (37.2)	1406 (39.0)	234 (29.3)	
Employment Status				<0.001
In paid workforce	3798 (86.2)	3169 (87.9)	629 (78.7)	
Not in paid workforce	605 (13.7)	436 (12.1)	169 (21.2)	
Living Arrangement				<0.001
Living alone	467 (10.6)	362 (10.0)	105 (13)	
Single parent with 1+ children	415 (9.4)	321 (8.9)	94 (11.8)	
Single with friends/relatives	268 (6.1)	198 (5.5)	70 (8.8)	
Couple with no children	753 (17.1)	622 (17.3)	131 (16.4)	
Couple with 1+ children	2471 (56.1)	2085 (57.8)	386 (48.3)	
Other ^2^	11 (0.2)	6 (0.2)	5 (0.6)	
Self-Reported General Health ^3^				<0.001
Poor/Fair	650 (14.8)	393 (10.9)	257 (32.2)	
Good	1701 (38.6)	1387 (38.5)	314 (39.3)	
Very Good/Excellent	2029 (46.1)	1806 (50.1)	223 (27.9)	

^1^ Total may differ due to missing data. ^2^ Alternate living arrangement (e.g., couple residing with nephew). ^3^ Groups were created by collapsing poor and fair responses (fair), and very good and excellent responses (excellent). ^4^ Chi-square tests of homogeneity between each variable and distress level group.

**Table 3 ijerph-19-14052-t003:** Physical activity and mastery level of participating sample at each timepoint.

	Wave 1 n = 4404		Wave 2 n = 4072		Wave 3 n = 3984		Wave 4 n = 3804		Wave 5 n = 3093	
Distress Level	Low	Elevated	Low	Elevated	Low	Elevated	Low	Elevated	Low	Elevated
	n (%)		n (%)		n (%)		n (%)		n (%)	
Survey non-participation ^1^	0 (0)	0 (0)	261 (7.2)	71 (8.9)	350 (9.7)	70 (8.8)	420 (11.7)	96 (12.0)	784 (21.7)	193 (24.2)
Physical Activity (weighted mins/week) ^2^										
0 to 59	618 (17.1)	218 (27.3)	504 (14.0)	212 (26.5)	553 (15.3)	192 (24.0)	553 (15.3)	192 (24.0)	327 (9.1)	135 (16.9)
60 to 149	578 (16.0)	151 (18.9)	561 (15.6)	162 (20.3)	497 (13.8)	117 (14.6)	497 (3.8)	117 (14.6)	415 (11.5)	107 (13.4)
150 to 300 ^3^	818 (22.7)	147 (18.4)	678 (18.8)	138 (17.3)	701 (19.4)	170 (21.3)	701 (19.4)	170 (21.3)	586 (16.3)	118 (14.8)
More than 300	1564 (43.4)	277 (34.7)	1538 (42.7)	212 (26.5)	1327 (36.8)	201 (25.2)	1327 (36.8)	201 (25.2)	1198 (33.2)	154 (19.3)
Mastery ^4^										
Low	-	-	833 (23.1)	545 (68.2)	843 (23.4)	574 (71.8)	840 (23.3)	535 (67.0)	758 (21.0)	416 (52.1)
High	-	-	2446 (67.9)	166 (20.8)	2379 (66.0)	143 (17.9)	2231 (61.9)	137 (17.1)	1780 (49.4)	108 (13.5)

^1^ Participant did not return questionnaire, returned questionnaire without data, deceased, withdrawn as not interested, incapable of response or temporarily unavailable/overseas. ^2^ Weighted minutes to account for higher intensity of vigorous physical activity. ^3^ Comparable with international guidelines. ^4^ Mastery not assessed at wave 1. Score dichotomized based on a one third vs. two third split to accommodate for a negatively skewed distribution: (1) Low, score of 25 or less, (2) High, score over 25.

**Table 4 ijerph-19-14052-t004:** Generalized estimating equations model of mastery and physical activity adjusted for sociodemographic and health characteristics ^1^.

Variables	B (SE)	Odds Ratio	95% Confidence Interval	
			Lower	Upper
Physical Activity (weighted mins/week) ^2^				
0 to 59	Ref			
60 to 149	−0.145 (0.088)	0.865	0.728	1.027
150 to 300 ^3^	−0.215 (0.091)	0.806	0.675	0.964
More than 300	−0.291 (0.095)	0.748	0.621	0.900
Mastery ^4^				
Low	Ref			
High	−2.069 (0.070)	0.126	0.110	0.145
Study wave				
2009	Ref			
2011	0.077 (0.038)	1.080	1.002	1.164
2013	0.051 (0.055)	1.052	0.944	1.172
2016	0.111 (0.085)	1.118	0.946	1.320
Gender				
Men	Ref			
Women	0.233 (0.091)	1.262	1.056	1.508
Education				
School	Ref			
Certificate/Diploma	−0.230 (0.109)	0.794	0.641	0.983
Bachelor/Postgraduate	−0.343 (0.106)	0.710	0.577	0.873
Self-Reported General Health ^5^				
Fair/Poor	Ref			
Good	−0.732 (0.084)	0.481	0.410	0.564
Very Good/Excellent	−1.141 (0.103)	0.243	0.199	0.298

^1^ Subgroups of predictors are compared for trend of distress during mid-age adulthood. Distress was measured over time and categorized into low or elevated distressed using Group-Based Trajectory Modelling. The reported results are from generalized estimating equations (GEE) analysis of distress group over a the 9-year study period. ^2^ Weighted minutes to account for time spent in vigorous intensity physical activity. ^3^ Comparable with international guidelines. ^4^ Dichotomized based on a one third vs. two third split to accommodate for a negatively skewed distribution: (1) Low, score of 25 or less, (2) High, score over 25. ^5^ Groups were created by collapsing poor and fair responses (fair), and very good and excellent responses (excellent).

## Data Availability

The data that support the findings of this study are not publicly available but are available on reasonable written request: a written request can be submitted to the corresponding author.

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
