# Peer review of "Longitudinal Association of Physical Activity, Mastery and Psychological Distress in Mid-Aged Adults over 9-Years"

_ijerph, 2022, doi:10.3390/ijerph192114052_

Round 1

Reviewer 1 Report

I do have some mild comments related to this study and shown as follows:

1.     Page 5 line 179, “Generalized estimating equations (GEE) regression analysis” was incorrect. It should be “Generalized estimating equations (GEE) method’s logistic regression”.

2.     In Table 2, the chi-square test should be used to test the homogeneity of middle-aged adults at baseline between two distress levels.

3.     The title of Table 4 “Generalized estimating equations model of mastery and physical activity with interaction adjusted for sociodemographic and health characteristics” was inconsistent with the description in the manuscript page 12 line 259 to 261.

4.     In Table 4, the word “Ref” should be removed in “Odds Ratio” and 95% CI.

Reviewer 2 Report

Longitudinal association of physical activity, mastery and psychological distress in Mid-aged adults over 9 years

Many thanks for having me to review the manuscript titled “Longitudinal association of physical activity, mastery and psychological distress in Mid-aged adults over 9 Years”. The authors drew attention to the benefits of physical activity and mastery of psychological distress of middle age adults via nine years five waves survey analysis. I appreciate the authors bringing solid evidence on the positive effects of regular physical activity and high mastery in mid-age adults’ well-being with abundant literature support and clear structural presentation. For this manuscript, I suggest the authors have a tiny revision before moving toward publication. Please see my comments to the authors below. 

For the abstract and introduction, the authors summarized the key processes and main findings and well presented the background and key terms of the current study. I was only a bit confused about the “morbidity” (line 10 and 19), whether the authors meant mortality because morbidity refers to the rate of prevalence.

As to the methodology, researchers achieved a significant sample size, followed up nine years with valid measurements. If possible, the authors may need to provide a clearer classification for the flowchart, since from wave 2 to wave 5, there were numbers of non-response but not withdrawn. The authors need to explain what differences between non-response and withdrawn participants. And if participants did not provide survey responses how and why researcher still kept them. Although researchers deal with missing data in the result section, around accumulative 50% of data missing possibly could impact the reliability and validity of results.

In addition, the Active Australia Survey fits the Australian population well and with good reliability, while participants only reported the activity in the last week (line 134). Researchers might better illuminate how the data can be generalised to the nine-year longitudinal study.

For discussion, the authors presented plentiful research but possibly need more analysis of how the literature related to the current study and results, such as lines 305 – 310 and 322 –323 the authors only listed the findings of previous research but did not elucidate the reasons and relationship to this study. Further, authors may better provide more explanation about the reasons to take physical activity and mastery as key variables in this study, whether a positive influence between physical activity and mastery exists, or other potential interactional effects function, although many studies have demonstrated the separately influences on adults’ wellbeing. 

 Overall, the authors well-structured and carefully presented their ponderable results, but only need a bit more explanation and illustration to publish level.

Possible useful references

Takeda F, Noguchi H, Monma T, Tamiya N (2015) How Possibly Do Leisure and Social Activities Impact Mental Health of Middle-Aged Adults in Japan?: An Evidence from a National Longitudinal Survey. PLOS ONE 10(10): e0139777. https://doi.org/10.1371/journal.pone.0139777

Abraham Cottagiri, S., Villeneuve, P. J., Raina, P., Griffith, L. E., Rainham, D., Dales, R., Peters, C. E., Ross, N. A., & Crouse, D. L. (2022). Increased urban greenness associated with improved mental health among middle-aged and older adults of the Canadian Longitudinal Study on Aging (CLSA). Environmental Research, 206, 112587. https://doi.org/10.1016/j.envres.2021.112587

Raina, P., Wolfson, C., Griffith, L. et al. A longitudinal analysis of the impact of the COVID-19 pandemic on the mental health of middle-aged and older adults from the Canadian Longitudinal Study on Aging. Nat Aging 1, 1137–1147 (2021). https://doi.org/10.1038/s43587-021-00128-1

Reviewer 3 Report

This excellent study examined the relationships between several variables and self-perceived psychological distress. This paper shows that mastery and physical activity as contributory factors do not interact. Particular strengths of the study were the large sample size and the longitudinal design (nine years). I believe that the statistical methods employed are appropriate and have been implemented appropriately and conclusions are consistent with the evidence. The the references are appropriate and I have no additional comments concerning the table and the figures. Two distinct longitudinally stable trajectories (high distress and low distress) were identified. In most instances the results presented here were consistent with prior literature. Specifically, a greater portion of individuals with elevated distress endorse poor self-rated health, were not in paid employment and had only secondary education. Women had greater representation in the high distress group.

Findings unique to this study

While both mastery and physical activity were associated with low distress, as previously observed, this contribution reports that there was no significant interaction between mastery and physical activity. Also, these authors found no significant relationship, between lower levels of physical activity and psychological distress. They note, however, that they assessed duration rather than frequency of activity which may account for the divergence from prior literature.

Can distinct populations of high distress and low distress be identified in this population? Do mastery and physical activity contribute to lower distress in adults? Do these factors interact? Is the level of distress longitudinally stable?
